# Measures of Corticalization

**DOI:** 10.3390/jcm11185463

**Published:** 2022-09-16

**Authors:** Marcin Kozakiewicz

**Affiliations:** Department of Maxillofacial Surgery, Medical University of Lodz, 113 Żeromskiego Str., 90-549 Lodz, Poland; marcin.kozakiewicz@umed.lodz.pl; Tel.: +48-42-6393422

**Keywords:** dental implants, long-term results, long-term success, functional loading, peri-implant bone, intra-oral radiographs, radiomics, texture analysis, corticalization, bone remodeling

## Abstract

After the insertion of dental implants into living bone, the condition of the peri-implant bone changes with time. Implant-loading phenomena can induce bone remodeling in the form of the corticalization of the trabecular bone. The aim of this study was to see how bone index (BI) values behave in areas of bone loss (radiographically translucent non-trabecular areas) and to propose other indices specifically dedicated to detecting corticalization in living bone. Eight measures of corticalization in clinical standardized intraoral radiographs were studied: mean optical density, entropy, differential entropy, long-run emphasis moment, BI, corticalization index ver. 1 and ver. 2 (CI v.1, CI v.2) and corticalization factor (CF). The analysis was conducted on 40 cortical bone image samples, 40 cancellous bone samples and 40 soft tissue samples. It was found that each measure distinguishes corticalization significantly (*p* < 0.001), but only CI v.1 and CI v.2 do so selectively. CF or the inverse of BI can serve as a measure of peri-implant bone corticalization. However, better measures are CIs as they are dedicated to detecting this phenomenon and allowing clear clinical deduction.

## 1. Introduction

The functional loading of dental implants induces permanent changes in the alveolar crest [1,2]. The functional loading of intraosseous dental implants causes significant changes in the structure of the alveolar marginal bone, observed radiographically [3]. There was corticalization and associated marginal bone loss relentlessly progressing over the five and ten years of observation presented previously [3]. It is expressed in the loss of trabeculation (lower entropy of bone radiostructure) in favor of the unification of the arrangement of bone components and their massification (increase of long elements in radiograph). Both of these structural changes are summarized in the bone index (BI). The conducted analysis strongly suggests that the phenomenon of corticalization is a nonbeneficial alteration of the bone around the implants (at least in the scope disclosed in this study). It means that marginal bone loss will increase as corticalization progresses.

The trabecular structure disappears and is successively replaced by cortical bone-like tissue. These observations were made on the digital analysis of peri-implant bone structure in intraoral radiographs. For this, the bone index [4,5] was used, or, strictly speaking, the inverse of this index since BI is used to detect trabecular bone. Due to the fact of dichotomous deduction possibilities (cancellous bone vs. cortical bone), 1/BI was proposed for detecting corticalization.

However, BI is oriented toward detecting cancellous bone. In trabecular structure radiographs, BI reaches the highest values. In contrast, it reaches low values in other bone structures. The author suspects that low BIs occur not only in images of cortical and corticalized bone but also in areas of bone atrophy (uniformly radiologically translucent). This suspicion is related to the structure of the BI [4,6] since there is a measure in its structure that highlights the existence of long strings of pixels of similar brightness (in other words, of similar radiographic translucency). This measure is not related to high brightness (optical density) only but shows both high and low optical density regions. BI cannot be the only measure for evaluating bone at dental implant site. Unfortunately, BI does not indicate whether a clinically suspicious site is corticalization or bone loss. This is a significant disadvantage. To avoid it, each examined site or radiograph should be subjected to visual inspection, which precludes the automation of the analysis and completely excludes the use of radiomics.

The aim of this study was to see how BI behaves in areas of bone atrophy (radiologically translucent non-trabecular areas) and to propose other indices specifically dedicated to detecting corticalization in living bone.

## 2. Materials and Methods

The source of the scientific material included in this study was digital intraoral radiographs [7] taken with the Digora Optima system (Soredex, Helsinki, Finland): 7 mA, 70 kV an 0.1 s (Focus apparatus—Instrumentarium Dental, Tuusula, Finland). Positioner Rinn (Densply, Charlotte, NC, USA) was used for the 90° angle of X beam to the surface of phosphor plate. Storage phosphor plates were read immediately after exposure.

Square areas of 3844 pixels (62 × 62), i.e., regions of interest (ROIs) in 8-bit, greyscale images were included in the study, numbering 40 for the compact (cortical) bone images, 40 for the cancellous (trabecular) bone images and 40 soft tissue images (Figure 1). A total of 120 ROIs were analyzed.

This provided information on three unique regions: cortical bone, trabecular bone and soft tissue. The textures of the X-ray images were analyzed in MaZda 4.6 freeware invented by the University of Technology in Lodz [8] to test measures of corticalization in control environments of trabecular bone (representing original bone before implant-dependent alterations) and soft tissue (representing product of marginal bone loss). MaZda provides both first-order (mean optical density) and second-order (entropy, differential entropy (DifEntr), long-run emphasis moment (LngREmph)) data. Due to the fact that the second-order data are given for four directions in the image, and in the present study, the author did not wish to search for directional features, the arithmetic mean of these four primary data was included for further analysis. The regions of interest (ROIs) were normalized (*μ* ± 3*σ*) to share the same average (*μ*) and standard deviation (*σ*) of optical density within the ROI. To eliminate noise, [9] worked on data reduced to 6 bits. For analysis in a co-occurrence matrix, a spacing of 5 pixels was chosen. In the formulas that follow, *p*(*i*) is a normalized histogram vector (i.e., histogram whose entries are divided by the total number of pixels in ROI), *i* = 1,2,..., *N_g_* denotes the number of optical density levels. Mean optical density (only a first-order feature) is calculated as follows:(1)$$\mu =\sum_{i=1}^{N_{g}}ip\left(i\right)$$

Second-order features are found by:(2)$$Entropy=-\sum_{i=1}^{N_{g}}\sum_{j=1}^{N_{g}}p\left(i,j\right)\log\left(p\left(i,j\right)\right)$$
(3)$$DifEntr=-\sum_{i=1}^{Ng}p_{x-y}\left(i\right)log\left(p_{x-y}\left(i\right)\right)$$
where Σ is sum, *Ng* is the number of levels of optical density in the radiograph, *i* and *j* are optical density of pixels 5-pixel distant one from another, *p* is probability and *log* is common logarithm [10]. The differential entropy calculated in this way is a measure of the overall scatter of bone structure elements in a radiograph. Its high values are typical for cancellous bone [4,11,12]. Next, the last primary texture feature was calculated (Figure 2):(4)$$LngREmph=\frac{\sum_{i=1}^{Ng}\sum_{k=1}^{Nr}k^{2}p\left(i,k\right)}{\sum_{i=1}^{Ng}\sum_{k=1}^{Nr}p\left(i,k\right)}$$
where *Σ* is sum; *Nr* is the number of series of pixels with density level *i* and length *k*; *Ng* is the number of levels for image optical density; *Nr* is the number of pixel in the series; and *p* is probability [13,14]. This texture feature describes thick, uniformly dense, radio-opaque bone structures in intraoral radiograph images [4,12].

The equations for DifEntr and LngREmph were subsequently used for the index construction [4,5,6,12]. The bone index (BI), which represents the ratio of the diversity of the structure observed in the radiograph to the measure of the presence of uniform longitudinal structures, was calculated:(5)$$Bone\;Index=\frac{DifEntr}{LngREmph}$$

Two more formulas were developed with the intention that they would describe the intuitive increases in their values together with the progression of corticalization and that they would suppress the results for cancellous and soft-tissue sites by representing such sites with low values:(6)$$Corticalization\;Index\;ver.1=\frac{LngREmph\cdot Mean}{DifEntr}$$
(7)$$Corticalization\;Index\;ver.2=\frac{LngREmph\cdot Mean}{Entropy}$$

The Kruskal–Wallis test was used for the comparison of medians between cortical, and trabecular or soft tissue radiograph (Statgraphics–StatPoint Technologies, Inc., The Plains, VA, USA). Factor analysis was used to find the statistically supported next measure for the corticalization process product. Input vectors: mean optical density, texture entropy, DifEntrp, and LngREmph. The procedure was performed for factors of eigenvalue ≥ 1. A probabilistic neural network (PNN) to classify cases into different ROI was applied. Rate of correctly classified ROIs by the network was evaluated.

## 3. Results

Calculations for selected measures of corticalization in radiographs of three types of tissue representing the corticalization phenomenon, bone loss and the reference region of cancellous bone are shown in Table 1. The results of the primary bone imaging features are shown in Figure 3. These consisted of one first-order feature (mean optical density) and two second-order features (DifEntr and LngREmph).

The constructed indices were then examined in three ROIs: bone index (Figure 4), corticalization index ver. 1 (Figure 5) and corticalization index ver. 2 (Figure 6).

The purpose of the factor analysis was to obtain a small number of factors that would account for most of the variability in the four textural features (mean optical density, texture entropy, DifEntrp, and LngREmph). In this case, the factor was extracted with a high eigenvalue, 3.30 (much greater than or equal to 1.0). It accounted for 82.4% of the variability in the original texture data. Since principal components was selected, the initial communality estimates were set to assume that all of the variability in the data was due to common factors. Moreover, the Kaiser–Meyer–Olkin measure of sampling adequacy (KMO) was above 0.6 for that set of input features. This factorability test indicates whether or not it is likely to be worthwhile attempting to extract factors from a set of variables. The KMO statistic provides an indication of how much common variance is present. For factorization to be worthwhile, KMO should normally be at least 0.6. Since KMO = 0.768, factorization was likely to provide interesting information about any underlying factors. The equation that estimated the common factor (the corticalization factor, CF) was performed to represent the factor loadings:(8)CF=0.8446·Mean+0.9555·Entropy+0.9066·DifEntr−0.9211·LngREmph
where the values of the variables in the equation are standardized by subtracting their means and dividing by their standard deviations. It also shows the estimated communalities, which can be interpreted as estimating the proportion of the variability in each variable attributable to the extracted factors.

Factor analysis indicated that by placing the main emphasis on the simple measurement of mean optical density and measuring the amount of chaoticity in the texture, it is possible to more than adequately detect corticalization sites in the bone image (high CF = 114 ± 23) with simultaneous indication of normal trabecular structure (intermediate values 80 ± 12) and sites that are no longer bone, such as those affected by marginal bone loss (lower values 29 ± 14). The presence of pixel long series of similar optical density is minimized in this corticalization evaluation technique. However, removing LngREmph from the analysis lowers the KMO to 0.618. Thus, one should suspect that short pixel series (i.e., the inverse of LngREmph) is more important in assessing corticalization. A second conclusion from this relationship is the essentiality of evaluating pixel series for indicating corticalization sites.

Thus, the last of the corticalization measures examined here was obtained from factor analysis: CF (Figure 7). It was strongly stratified and allowed for good discrimination of cortical bone from cancellous bone, cancellous bone from soft tissue, and soft tissue from cortical bone (*p* < 0.001).

The relationships of the corticalization index with the bone index and corticalization factor are shown in Figure 8 below.

## 4. Discussion

It is worth noting that a simple measure derived from radiograph histogram analysis (i.e., mean optical density) has been used in dentomaxillofacial radiology for decades [9,15,16,17,18,19,20,21,22,23,24]. It carries a great deal of clinically useful information but does not allow the automation feature of radiomics [25,26,27,28] because it requires an analog context for understanding the significance of local density changes. A second argument is the vast amount of other information coming from the radiograph beyond the histogram data. In dentistry, more than 10,000 texture features computing from the determined ROIs are now possible [29]. A final issue is the non-specificity of the mean optical density for assessing bone corticalization since residual granules of biomaterial previously implanted into bone, for example, can be detected by this feature [30,31].

In peri-implant bone, optical density increases on plain intraoral radiography in patients treated with immediate-loading implants [1,32]. Similar observations were made for late-loaded implants in the same time horizon (12 months of functional loading) that were noted in the peri-implant bone texture structure [33]. The sum of squares, a feature from the co-occurrence matrix, was studied, and it was found that there is a significant decrease in the value of this texture feature around the integrated dental implant at 12 months after prosthetic loading. This indicates a homogenization of the bone texture and a decrease in its intrinsic contrast [33]. These are the initial reports describing the phenomenon of corticalization of the alveolar crest caused by dental implants.

How to measure the product quantity derived from corticalization process in peri-implant bone in a clinical situation is a critical question. In recent years, the occurrence of corticalization in peri-implant bone was mentioned in scientific literature [32,34,35,36], and attempts are being made to describe this phenomenon [3] and get to know its clinical significance.

The importance of the standardization of images and ROI should be emphasized first. The approach to this issue depends on the tools used later. When digital radiographic subtraction is used [7,9,16,17,18,21,22], geometric alignment is necessary first because two radiographs of the same implant but taken at different times are superimposed. Rotational, translational, scale and affine distortions need to be corrected. Next, alignment and contrast brightness are needed. This is best achieved by aligning the histograms of the reference locations [20,23]. In the research presented in this study, this second range of alignment is essential. An alignment algorithm is introduced in the MaZda program consists of standardizing the ROI. The ROIs were adjusted (μ ± 3σ) to share the same average (μ) and standard deviation (σ) of optical density within the ROI.

Bone index (BI) is a good measure for determining the qualitative changes occurring in the cancellous bone of the jaw. A decrease in its value indicates the disappearance of the structure characteristic of trabecular bone. This is most likely related to corticalization since low BI precedes the appearance of marginal trabecular bone in dental implantology by years. There was a strong association of low BI (0.41 ± 0.19) present in the fifth year of implant use with marginal bone loss at that time (*p* < 0.0001) [3]. Bone index also well describes the results of guided bone regeneration inside the alveolus [4]. It highlights the appearance of bone trabeculae at the site of biomaterial implantation. Inferences based on the BI also seem to work well in other disciplines, i.e., in the analysis of bone consolidation quality [37], where a low BI is present at post-fracture sites because the islands of bone densities are more homogeneous (compact) than cancellous bone. This results in a less-chaotic structure, i.e., entropy is reduced (and this entropy is the numerator of the fraction forming the BI). For the same reasons, the bone image here has broad and uniform radio-opaque fields, where long lines of pixels of the same optical density can be found. This causes a high LngREmph, and this is the denominator of the fraction that forms BI. Thus, it affects the reduction of the final BI to 0.70–0.79. Unfortunately, LngREmph (as well BI) cannot describe the pixel series as high optical density (bone apposition) or low optical density (bone loss) points in an image (BI is also low in homogenous radiologically translucent regions).

Moreover, in a study of corticalization, one would need a tool indicating the sites of corticalization rather than the inverse, a tool indicating trabeculation, since bone loss is much more strongly represented in this index (i.e., 1/BI) than corticalization itself. Corticalization index ver. 1 is based on up to the inverse of the described bone index enhanced by the mean optical density result in the ROI. Corticalization index ver. 2 in its design is similar to the inverse of the texture index described previously [5].

The presence of statistically significant differences between the ROIs of the corticalization measures studied indicates that they are all useful to some degree. However, the three tissues tested differed from each other in mean optical density, differential entropy, bone index, corticalization factor). This does not provide a simple measure indicating the site of corticalization searched for in this study. Nevertheless, it should be stated here that both mean optical density and corticalization factor are highest at the site of corticalization. Entropy, on the other hand, is uniformly elevated in both bone tissues (i.e., corticalized and trabecular) relative to soft tissue, which is almost lacking a chaotically scattered texture pattern (Table 1). The next one-element group is the long-run emphasis moment (LngREmph), whose value is highest in soft tissue and lowest in both bone types studied (*p* < 0.001). Yet other measures studied here (both corticalization indices) unambiguously indicate that a given site is corticalized and significantly different from both trabecular bone and well bone loss (i.e., soft tissue). Thus, these two measures do not have the interpretive contamination of random bone loss detection introduced inside. High values here indicate only a corticalization phenomenon, in contrast to the low values, which indicate everything else, i.e., trabecular bone and bone loss. Therefore, these two measures also cannot be used for evaluating the results of guided bone regeneration (the bone index is great for this purpose), nor can they be used to study sites of bone loss. It will be possible in the future to select the best measure for studying a particular phenomenon in peri-implant bone. However, for the considerations in this study, the indices of corticalization (CI ver.1 and CI ver. 2) are the most interesting measures selected.

It seems appropriate to present a dedicated index for detecting corticalization as a phenomenon important for the long-term success of implant treatment. The purpose of this paper is to present such an index for clinical application. The interpretation inconvenience of 1/BI can be resolved by associating the index with high brightness (i.e., high optical density) as typically found in radiographic images of compact (cortical) bone. This gave the idea to include in the index the first-order feature of mean brightness/optical density, i.e., the mean from the histogram of the examined region of interest.

Both versions of CI allow for distinguishing corticalization from atrophy and, of course, also from trabecular bone. The question is whether it is better to rely on BI in the assessment of corticalization, which after all is designed to search for trabecular bone in X-ray images and which also distinguishes bone atrophy well (BI for cancellous bone is the highest, for cortical bone, it is statistically significantly lower and for the site of bone loss, it is significantly lower to corticalized bone). Or is it better to rely on a CI that indicates only corticalization and describes trabecular bone and bone loss together at an equal level i.e., CI ver.1 approx. 114, CI ver.2 approx. 52 for non-corticalization sites.

When considering these aspects of the analysis, it should be emphasized that marginal bone loss has already been very well described in the literature, and the methods to diagnose it are known and evident [38,39,40,41,42,43]. It is also known to be an unequivocally unfavorable phenomenon for implant success [44,45,46]. Corticalization itself is suspected as a potentially unfavorable prognostic factor [3]. Thus, it seems best at this stage of the understanding of this phenomenon to focus on the dichotomous separation of the peri-implant bone image structure: corticalized versus other (i.e., cancellous or affected by loss).

As is well known, cortical structural changes can have very serious adverse effects, as in patients treated with antiresorptive therapy (MRONJ) [47], less severe as it seems after a decade of monitoring bone transformation around dental implants [3] or perhaps have a positive effect as in the case of immediate loading of dental implants [32].

The phenomenon described here is so pronounced in the jaws because the bone appositional index here is one of the highest in the body [48]. It is certainly higher than in the iliac bone, femur or vertebrae. In the mandible, the bone apposition rate is 0.086–0.088 µm per hour. This process guarantees the osseointegration of the dental implants in the first phase but is probably responsible in later years for the corticalization of the surrounding area modulated by the permanent loading of the bone by biting and chewing forces. The remodeling and the superimposition of new osteons to the older [49]. This late effect at sufficiently high levels can lead to increased bone fragility and brittleness through the mechanism of osteon hypermineralization, which is related to the process of bone apposition [50]. However, in the jawbone, the phenomenon occurs on a microscale. Whether it is a negative process for the long-term maintenance of functioning dental implants is uncertain. Perhaps for this reason, it is worth thinking about other methods of building measures to describe bone corticalization.

A new approach is the use of factor analysis to evaluate peri-implant bone. The hopes placed in it are based on good predictive experiences from other surgical teams [51,52]. The advantage is the statistically reliable combination of information flowing from several texture features into a single number (corticalization factor) describing the variation occurring in, for example, four features: mean optical density, the frequency of long series of pixels of similar optical density (LngREmph) and two measures of texture pattern scattering (entropy and differential entropy). This aforementioned reliability relies on high eigenvalues and KMO statistics.

Information describing the triplet variation (cortical vs. trabecular vs. bone loss) can reduce the number of features while retaining their internal information by the factor analysis and equation for the corticalization factor (8). It turns out that based on the factor calculated in this way (CF), statistically significant differences (*p* < 0.05) can be indicated between corticalization site roentgenograph (value approx. 120) and cancellous bone (value about 80) versus bone loss (value less than 40). Both analyses (Figure 8) indicate the possibility of the direct transition of cancellous bone to atrophy and of cortical bone to atrophy. Based on the presented methods for the detection of corticalization, there is no indication of a transition state between cancellous bone and bone loss. It is certainly not a corticalized bone. Bone loss can arise directly from one or the other bone tissue.

On the other hand, when considering the interpretive convenience in a study of only corticalization, it is more comfortable to use a tool that gives dichotomously differentiated results, i.e., yes or no for corticalization, and such measures are presented above: the corticalization indices.

It is important and interesting to validate the corticalization measures presented here on a wide range of patients in variable clinical situations (e.g., different implants [53,54], different surgical protocols [55], bone compression screws [56,57], different prosthetic work [58], vitamin D3 levels [47] and molecular signaling modulation [59]). This will allow for choosing the best applications for particular measures. Or perhaps it will prove advisable to use several measures simultaneously, e.g., for monitoring remodeling of the cancellous and cortical substance. It is also important to test the usefulness in other parts of skeletal surgery such as hand [60], foot [61], thoracic [62], orthognathic [63], spine [64], joint replacements [35,65] and rehabilitation [66,67].

One should not forget other measures of corticalization being developed in dentistry itself like fractal dimension and multifractal spectra [53,68,69,70]. It seems that this valuable and interesting source may in the future bring very useful measures of peri-implant bone remodeling.

This study provides important clinical considerations for dentistry (especially dental implantology). First, it systematizes the possibilities of assessing bone remodeling. It will be possible in the future to select the most suitable index in relation to the observed bone remodeling processes. Second, it is important to relate an objective measure of bone condition with the prediction of dental implant maintenance in normal function. The proposed measures of corticalization are applicable for monitoring bone health around the dental abutments associated with bridges and crowns and the results of guided bone regeneration and tissue bone regeneration in implantology and perioodontology. The clinical relevance of this study can also be seen in monitoring antiresorptive therapy in the treatment of osteoporosis and controlling the metastasis of malignant tumors to bone. Orthopedists, neurosurgeons and hand surgeons who also use metal stabilizers, screws, cages and joint replacements may also benefit. Many fields of medicine need evaluators that assess the condition of bone and its transformation as a function of time. In these example fields of medicine, for example, the use of the corticalization index is being looked at.

## 5. Conclusions

The corticalization factor, or inverse of bone index, can serve as a measure of peri-implant bone corticalization. However, better measures are the corticalization indices as these are dedicated to detecting corticalization and allowing for clear clinical deduction.

## Figures and Tables

**Figure 1 jcm-11-05463-f001:**
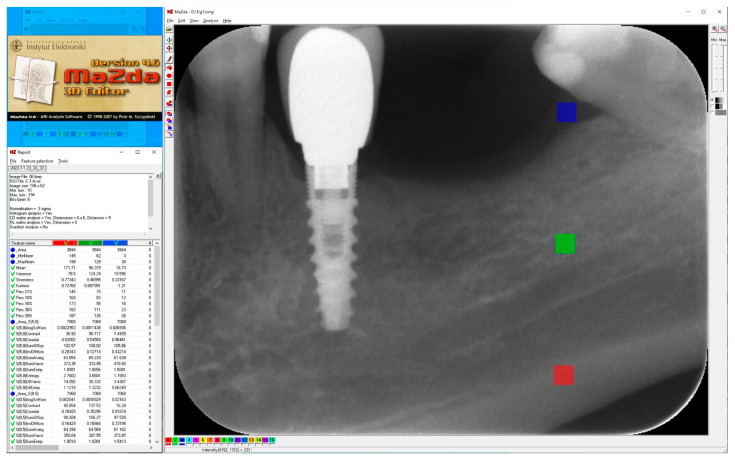
Regions of interest were located in cortical bone (ROI 1), trabecular bone (ROI 2) and soft tissue (ROI 3) in main window of MaZda. Next, a series of textural features was extracted (MZ Reports—on the left) and exported in comma-separated vector format (CSV).

**Figure 2 jcm-11-05463-f002:**
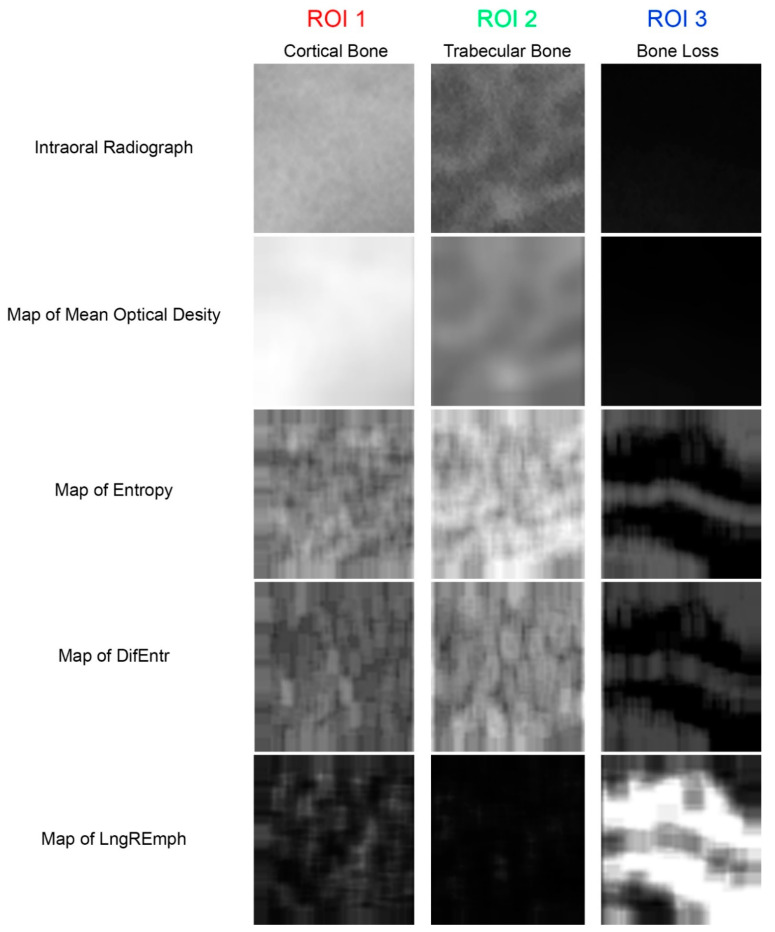
The source material and the primary texture features extracted from it. The meanings of the ROIs are the same as in Figure 1. Maps of the local intensity of the studied features are below the original radiographs. The map is created from square boxes of nine pixels. In the maps of features, lighter areas indicate higher local intensity of the feature, while darker areas indicate lower intensity of the feature.

**Figure 3 jcm-11-05463-f003:**
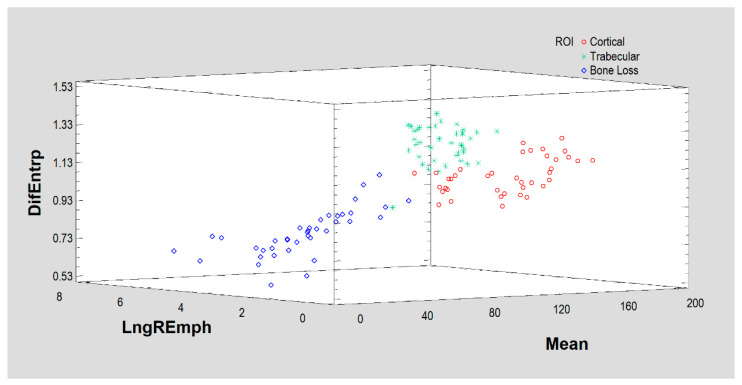
Based on the above three features (DifEntrp, Mean optical density, LngREmph), the algorithm manages to initially separate the results for the three tissues (ROIs), but corticalization (Cortical) is not well discriminated here. It is worth noting that the simple measure of mean optical density itself shows the differences between the regions of interest studied.

**Figure 4 jcm-11-05463-f004:**
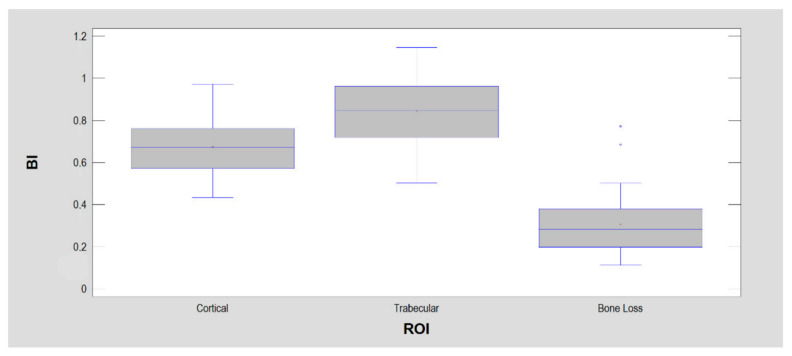
The bone index (BI) was calculated for the detection of normal bone (i.e., trabecular bone) within dental alveolus during guide bone regeneration. That is why BI reaches the highest values in ROI 2 representing trabecular bone. There are significant statistical differences between each ROI.

**Figure 5 jcm-11-05463-f005:**
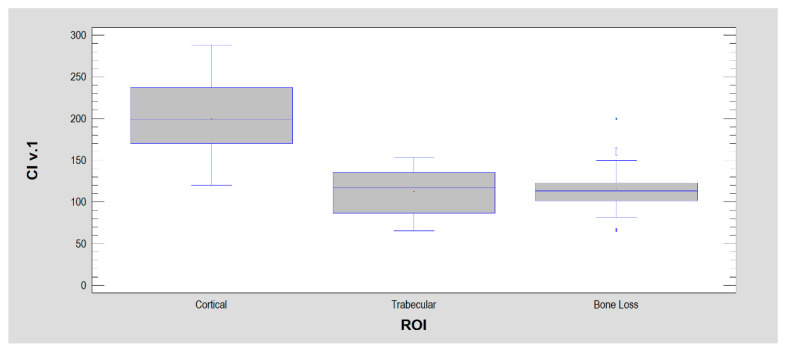
Corticalization index ver. 1 (CI v. 1) is based on two components included in BI and mean optical density. The components are arranged inversely to the BI to emphasize the corticalization sites rather than trabeculation, and the mean optical density enhances this effect because it is located in the numerator and is highest in the cortical bone.

**Figure 6 jcm-11-05463-f006:**
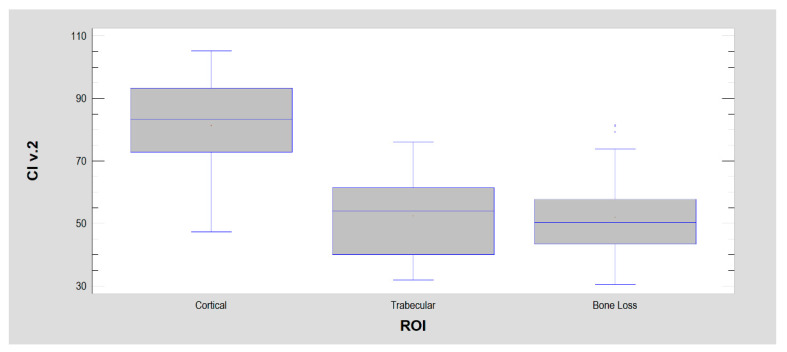
Corticalization index ver. 2 (CI v. 2). This corticalization measure differs from version 1 by replacing differential entropy (ver. 1) in the denominator with entropy (here, ver. 2). This was dictated by the good statistical separation of ROI 1 from the other two ROIs by entropy. However, due to the greater spread of entropy in ROIs than differential entropy, the separation between ROIs is weaker here (but still highly statistically significant: *p* < 0.001).

**Figure 7 jcm-11-05463-f007:**
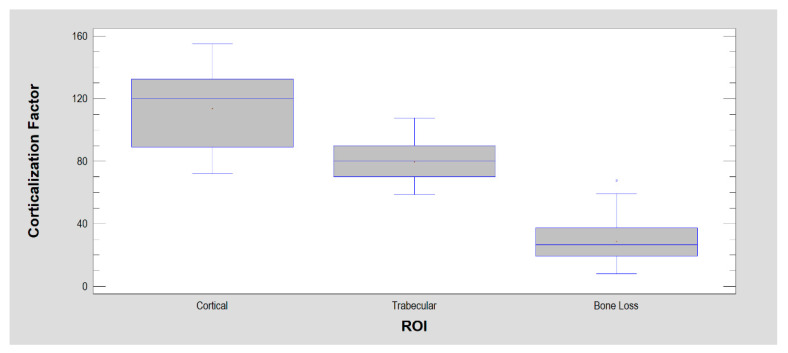
Corticalization Factor (CF). It has statistical features similar to BI, but it is most strongly expressed in cortical sites, weaker in trabecular bone and weakest in soft tissues.

**Figure 8 jcm-11-05463-f008:**
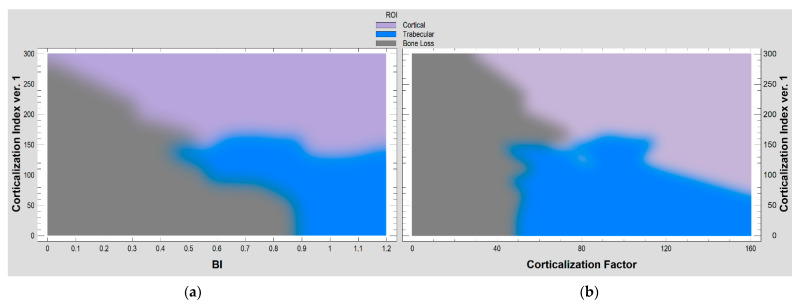
The relationships of selected corticalization measures in the evaluation of intraoral radiographs. (**a**) Corticalization index ver.1 with bone index (BI). A probabilistic neural network (PNN) used to classify cases into different three structures in radiograph (cortical, trabecular, bone loss), based on two input variables (corticalization index ver.1 and bone index). Of the 120 ROIs, 93% were correctly classified by the network. (**b**) Corticalization index ver.1 with corticalization factor. Among the 120 ROIs used, 94% were correctly classified in this pair of corticalization evaluators.

**Table 1 jcm-11-05463-t001:** Numerical results of the investigation of selected measures of corticalization.

Measure of Corticalization	ROI 1 Cortical Bone	ROI 2 Trabecular Bone	ROI 3 Bone Loss	Note
Mean Optical Density	132 ± 27	91 ± 15	34 ± 15	*p* < 0.001 ^1^
Entropy	2.68 ± 0.15	2.74 ± 0.19	1.79 ± 0.27	*p* < 0.001 ^2^
Differential Entropy	1.10 ± 0.09	1.28 ± 0.10	0.81 ± 0.15	*p* < 0.001 ^1^
LngREmph	1.66 ± 0.21	1.55 ± 0.18	3.01 0.97	*p* < 0.001 ^3^
Bone Index	0.67 ± 0.13	0.84 ± 0.15	0.31 ± 0.14	*p* < 0.001 ^1^
Corticalization Index ver.1	200 ± 42	112 ± 28	115 ± 26	*p* < 0.001 ^4^
Corticalization Index ver.2	81 ± 15	53 ± 13	52 ± 12	*p* < 0.001 ^4^
Corticalization Factor	114 ± 23	80 ± 12	29 ± 14	*p* < 0.001 ^1^

^1^ Statistically significant difference found between all the ROIs compared with each other; ^2^ ROI 3 is significantly lower than ROI 1 as well ROI 2; ^3^ ROI 3 is significantly higher than ROI 1 as well as ROI 2; ^4^ ROI 1 is significantly higher than ROI 2 as well as ROI 3. ROI—region of interest; LngREmph—long-run emphasis moment.

## Data Availability

The data on which this study is based will be made available upon request at https://www.researchgate.net/profile/Marcin-Kozakiewicz access on 30 June 2022.

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
