# Peer review of "Measures of Corticalization"

_jcm, 2022, doi:10.3390/jcm11185463_

Round 1
Reviewer 1 Report
Dear author,
The manuscript clearly comes from "What does bone corticalization around dental mean in light of 10 years follow up". It is interesting the approach you gave to measures of corticalization. I would like to recommend to explore more the BI lacks and clinical applicability/advantage of other index in Introduction.
Author Response
Kindly find my answer in the attached file

Reviewer 2 Report
Thank you for the opportunity to review this article.
#The title is too simple and boring. I suggest the authors to change or complete for a more creative title.
#Abstract and Intro: "It has been noted that after insertion of dental implants into living bone..." Remove "it has been noted" and rephrase.
#Introduction: "(dark non-trabecular areas)" please, change "dark" for the appropriate term in all the text of the article
#Introduction: "There is corticalization and associated marginal bone loss relentlessly pro- 26 gressing over the five and ten years of observation presented here...." Are those results? Please, clarify whose results are or move to the right section."
#Methods: Ethical committee number approval is missing.
#What is the clinical relevance of this study?, Please, add in the discussion
#How a dentist can use this calculation? I mean, it is too complicated to apply in the routine...
#Discussion: "It was noted increases..." please, rephrase
#Discussion, 4th phrase: "...peri-implant bone has been noted..." the article needs to a native English review to avoid repetitive use of inadequate words.
#Discussion is too long and boring. I suggest tho reformulate it with the objective of emphasize the applications of your work in the clinical assessment of the corticalization.
All the best!
Author Response

(The authors gave the same response as above.)

Reviewer 3 Report
The paper aimed to test new radiological indices to assess peri-implant bone changes. The proposal is interesting and the article describes in detail the methodology of the study and appropriately discusses its results. I suggest, however, including a discussion on the need to standardize the histogram of images from the same patient when the technique is used clinically to compare possible bone changes over time, since the density and contrast of the intraoral radiographic image are influenced. by several factors that can change the results of this evaluation.
Author Response

(The authors gave the same response as above.)

Reviewer 4 Report
The paper aims to find distinctive textural features of corticalization. A total of 120 square (62x62 pixels) ROIs were analyzed. Eight metrics were studied and Corticalization Index, Bone Index and Corticalization Factor were found as selective metrics.
My only concern is that the manuscript has excessive number of self citations (23/67). Author may reduce that ratio by excluding irrelevant self citations. English language and style are fine/minor spell check required.
Author Response

(The authors gave the same response as above.)
